# COVID-19 vaccine for people who live and work in prisons worldwide: A scoping review

**Nasrul Ismail**[1], **Lara Tavoschi**[2], **Babak Moazen**[3,4], **Alicia Roselló**[5], **Emma Plugge**[5,6]*

**1** School for Policy Studies, University of Bristol, Bristol, United Kingdom, **2** Department of Translational Research and New Technologies in Medicine and Surgery, University of Pisa, Pisa, Italy, **3** Department of Health and Social Work, Institute of Addiction Research (ISFF), Frankfurt University of Applied Sciences, Frankfurt/Main, Germany, **4** Heidelberg Institute of Global Health (HIGH), Heidelberg University, Heidelberg, Germany, **5** UK Health Security Agency, London, United Kingdom, **6** Faculty of Medicine, University of Southampton, Southampton, United Kingdom

* E.Plugge@soton.ac.uk

**Data Availability Statement:** All relevant data are within the paper and its Supporting information files.

## Abstract

Overcrowding, poor conditions, and high population turnover make prisons highly susceptible to COVID-19. Vaccination is key to controlling COVID-19, yet there is disagreement regarding whether people who live and work in prisons should be prioritised in national vaccination programmes. To help resolve this, we critically examine the extent, nature, and quality of extant literature regarding prioritisation of COVID-19 vaccinations for people who live and work in prisons. Using a scoping review as our methodological framework, we conducted a systematic literature search of 17 databases. From 2,307 potentially eligible articles, we removed duplicates and screened titles and abstracts to retain 45 articles for review and quality appraisal. Findings indicated that while most countries recognise that prisons are at risk of high levels of COVID-19 transmission, only a minority have explicitly prioritised people who live and work in prisons for COVID-19 vaccination. Even among those that have, prioritisation criteria vary considerably. This is set against a backdrop of political barriers, such as politicians questioning the moral deservingness of people in prison; policy barriers, such as the absence of a unified international framework of how vaccine prioritisation should proceed in prisons; logistical barriers regarding vaccine administration in prisons; and behavioural barriers including vaccine hesitancy. We outline five strategies to prioritise people who live and work in prisons in COVID-19 vaccination plans: (1) improving data collection on COVID-19 vaccination, (2) reducing the number of people imprisoned, (3) tackling vaccine populism through advocacy, (4) challenging arbitrary prioritisation processes via legal processes, and (5) conducting more empirical research on COVID-19 vaccination planning, delivery, and acceptability. Implementing these strategies would help to reduce the impact of COVID-19 on the prison population, prevent community transmission, improve vaccine uptake in prisons beyond the current pandemic, foster political accountability, and inform future decision-making.

**Funding:** EP, LT and BM are members of the Reaching the hard-to-reach: Increasing access and vaccine uptake among prison population in Europe (RISE-Vac) project. The RISE-Vac project is supported by an EU Health Programme (EHP-PJ-2020). The funder had no role in study design, data collection and analysis, decision to publish, or preparation of the manuscript.

**Competing interests:** The authors have declared that no competing interests exist.

## Introduction

In March 2020 the World Health Organization (WHO) declared the COVID-19 pandemic. At the time of writing, there have been 434,154,739 reported cases of COVID-19 and 5,944,342 deaths [1].

Concurrently, there are over 12 million people imprisoned globally [2]. People in prisons are more likely to experience physical and mental illness compared to the general public [3, 4]. The risk of criminality and imprisonment is heightened by the cumulative influence of political, economic, environmental, and social factors, such as poverty and social exclusion, in addition to individual lack of access to healthcare services and behavioural factors [5].

Labelled as 'epidemiological pumps' [6], prisons are highly susceptible to COVID-19 due to issues such as overcrowding, insalubrious and unsanitary conditions, suboptimal healthcare services, and high population turnover [7]. Highly transmissible respiratory pathogens can have devastating impacts on overcrowded prison populations, as shown by the historical examples of Spanish Flu in 1918 [8], and tuberculosis in Russian prisons in 2000 [9]. As of December 2021, there were an estimated 612,602 positive cases of COVID-19 among people in prisons, and 4,305 deaths [10]. The true scale of COVID-19 cases and deaths in prisons globally remains unclear because of significant underreporting or non-reporting by at least 148 countries [10]. The higher prevalence of chronic illnesses in imprisoned people puts them at greater risk of severe COVID or death than their peers in the community.

It is often difficult to enforce even the most rudimentary standards of COVID-19 mitigation in prisons. Social distancing is problematic because of overcrowding, which is estimated to occur in the prisons of 60 per cent of countries [11]. While it has been recommended that prisons provide access to clean water and enable hygiene practices, such as handwashing, isolation, and quarantine [12], these are often difficult to implement because of the prison architecture and infrastructure. Prisons also often have limited access to testing and personal protective equipment (PPE) [12]. Thus, disease control strategies in prisons often focus on restrictions of movement, which result in significant stress and anxiety for the imprisoned population [13, 14].

Discussions regarding the pandemic's impact on prisons often ignore the health impact on prison staff, a situation hindered by the lack of robust data on positive COVID-19 cases and deaths among these workers. Epidemiological studies have found that prison staff are more vulnerable to infectious diseases, with a higher prevalence of physical illness than the general population, including hepatitis, HIV, and tuberculosis than the general population [12, 15]. A higher burden of mental health concerns is also apparent [16]. These are often triggered by organisational and infrastructural issues such as staffing shortages, resulting in both presenteeism and absenteeism [17].

Prison officers have also been found to be at higher risk of COVID-19 infection compared to staff working in health and social care [18]. The oscillation of these individuals between prisons and the community also elevates risks of infection transmission for people in their care. Staff can act as a vector, unwittingly bringing infection into the prison [19, 20]. If infected, staff must undergo self-isolation, which can result in staffing pressures. This may in turn impact people living in prison through reduced time outside cells, meaning less access to activities that might ameliorate their anxiety, fear, and aggression [21, 22]. It may also impact the health and wellbeing of the remaining staff who are working.

Vaccination is key to controlling COVID-19 and reducing its impact on the population. Vaccines have been proven to reduce the prevalence of communicable diseases in prisons [23], and prison vaccination drives enable countries to fulfil existing legal obligations for prison health, such as the Mandela, Bangkok, Beijing, and Havana Rules [24–27]. Collectively, these

concordats demand recognition of the vulnerability of prison populations, as well as regard for the principle of equivalence and for governments' duty to protect those in their care whose liberty has been deprived. Article 12 of the International Covenant on Economic, Social and Cultural Rights realises "the right of everyone to the enjoyment of the highest attainable standard of physical and mental health", which applies to all individuals irrespective of their status under the law.

Nonetheless, people who live and work in prisons continue to be left behind by vaccination programmes [28]. While countries such as Spain, Poland, Finland, Ireland, and Sweden have vaccinated the majority of their prison population [29], many other countries have not. In the US, not all states have prioritised prisons in state vaccination plans [30]. Given emerging evidence that COVID-19 disproportionately affects prison populations resulting in widening health inequalities, prioritising incarcerated populations would accelerate vaccination opportunities to people who are often underserved by community health programmes, while simultaneously promoting the health and wellbeing of all individuals, irrespective of age, in line with Goal 3 of the UN's Sustainable Development Goals 2030 [31].

The aim of this scoping review is to critically examine the extent, nature, and quality of available literature on COVID-19 vaccinations for people who live and work in prisons. We intend to capture a comprehensive picture of the current debates on this topic, as well as identify relations, gaps, contradictions and inconsistencies in publications on this topic, all the time ensuring rigour and transparency in our research methods. Our analysis of the extant literature on this topic is followed by recommendations for politicians, policymakers, academics, and practitioners for developing policies and interventions which could increase vaccination rates in prisons such that they are in line with those in the wider community, alongside important directions for future research.

## Methodology

Our scoping review proceeded using the five-stage process outlined by Arksey and O'Malley [32]: 1) identifying the research question; 2) identifying relevant studies; 3) study selection; 4) charting the data; and 5) collating, summarising and analysing the included literature. The review is reported in accordance with the Preferred Reporting Items for Systematic Reviews and Meta-Analyses extension for Scoping Reviews (PRISMA-ScR). The review was exempt from institutional ethical approval as it did not involve human subjects.

### Identifying the research question

Our review was guided by the question 'What is the extent, nature, and quality of the literature on COVID-19 vaccinations for people who live and work in prisons?'. We define 'prison' as any setting that houses individuals aged 18 and above who have been deprived of their liberty because of a criminal conviction. We excluded other locations that house people deprived of their liberty, including police cells, military detention centres, immigration removal centres, and mental health institutions.

### Identifying relevant studies

We conducted a systematic literature search using 17 databases, reflecting the breadth of disciplines in this area of study: Applied Social Sciences Index & Abstracts (ASSIA), BioMed Central, Cumulative Index to Nursing and Allied Health Literature (CINAHL), Cochrane Library, Embase, EBSCO eBook Collection, MEDLINE, ProQuest Criminal Justice, PsycINFO, SAGE Journals, ScienceDirect, Scopus, Springer Link, Taylor & Francis, Web of Science, and Wiley Online Library.

We deliberately used broad search terms to maximise search sensitivity (S1 Appendix). We also used Boolean operators such as "OR" and "AND" to either combine or separate search terms, along with Medical Subject Headings (MeSH) terms. Our search was restricted to literature published between December 2019, when COVID-19 was discovered, and September 2021.

Our search of grey literature focused on official channels of information such as international and government sources, and professional bodies that champion the interests of people who live and work in prisons. To ensure comprehensive coverage of the literature, we also hand-searched reference lists of included studies.

## Study selection

We screened articles by title and abstract and obtained full text articles for potentially eligible studies. There were no restrictions on country or language of publication, and all study designs were accepted. We included papers that met the following eligibility criteria:

- Published in peer-reviewed journals or grey literature

- Published between December 2019 and September 2021

- Examined COVID-19 vaccination among people in prisons and/or prison staff

  Exclusion criteria were as follows:

- Publications that did not focus on prisons as a setting

- Publications where analysis of COVID-19 vaccination in prison is non-existent or marginal

While we did not exclude papers that were published before the availability of vaccines, most included studies were published after vaccine availability.

Two reviewers (NI and EP) initially screened 10% of the records independently. This involved reading and reviewing the articles in full, discussing the outcome of these reviews, and resolving any disagreements. Following this procedure, the reviewers proceeded with the full review of all articles.

We evaluated papers that reported research data using the National Institute of Health's quality assessment tools for quantitative research [33], and the Critical Appraisal Skills Programme checklist for qualitative research [34]. For publications that used cost-effectiveness analysis or mathematical modelling approaches, we used the Consolidated Health Economic Evaluation Reporting Standards (CHEERS) statement for our evaluation [35].

Two independent reviewers were assigned to appraise the quality of each included study using the following domains: clear statement of research aims, appropriateness of research question, research design, rigour of data collection and analysis, study findings, authors' acknowledgement of possible biases, ethical considerations, and value of the research. Any disagreements in assessment were resolved by a third independent reviewer.

We did not assess the quality of opinion pieces, case studies, or grey literature, as no robust analytical framework yet exists with which to assess the above-mentioned publication types.

## Charting, collating, summarising, and analysing the data

We entered key details into a table using the following headings: author, month and year, publication title, study design, population described or studied (including geographical coverage), key findings, and recommendations related to COVID-19 vaccinations for people who live and work in prisons. S2 Appendix provides the abbreviated data charting analysis.

Once familiar with the data of included studies, we used thematic content analysis to identify key themes. Following an inductive process, we identified the following six themes:

1. Epidemiological arguments on vaccinating people in prisons

2. The extent to which people in prisons are being prioritised for COVID-19 vaccination

3. Facilitators and barriers to COVID-19 vaccination uptake in prisons

4. Vaccine hesitancy

5. Vaccine populism

6. Access to vaccines as a human right

7. Vaccination should run in parallel with other mitigation actions

These themes were developed through an iterative process of review and refinement by the authors.

## Results

Our search returned a total of 2,307 articles. After removing duplicates and screening titles and abstracts, 112 articles were retained, all of which were reviewed in full against our inclusion and exclusion criteria. In total, 45 papers were included for data extraction. Fig 1 displays the flow of articles through the review process.

### Characteristics of included studies

Of the 45 publications judged eligible for inclusion, nearly seven in ten ($n$ = 31, 69%) were opinion papers. These were typically analysis papers, though some editorials and correspondences were also included. The remaining 14 papers (31%) were empirical studies. These included one retrospective cohort study [36], three cross-sectional survey [37–39], one genome sequencing study [40], one morbidity and mortality report [41], three mathematical modelling papers [42–44], four secondary data analysis papers [45–48] and one qualitative study [49].

Most included studies were published in countries of the Global North. More than half ($n$ = 25, 57%) were published in the United States [36, 37, 39–62]; five were published in the

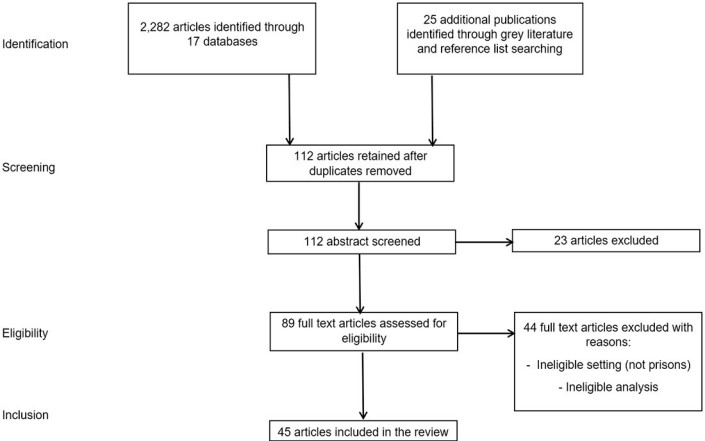

**Fig 1. Preferred reporting items for systematic reviews and meta-analyses flowchart.**

United Kingdom [63–67]; two were from Brazil [68, 69]; and there was one publication each from Africa [70], Canada [71–73], and South Korea [38].

An additional nine papers focused on various world regions. Three had a global focus [71, 73, 74], four focused solely on Europe [12, 75–77], and two presented comparative accounts, one between the UK, US, and Europe [78], and one between the US and Brazil [79]. S2 Appendix provides a breakdown on the countries and regions of focus for each of the pieces of literature. Countries and regions not listed in this document have not published data which prioritised people who live and work in prisons at the time of analysis.

The majority of empirical studies were judged to be of good quality (see S3 Appendix).

## Themes from included studies

**Epidemiological arguments on vaccinating people in prisons.** Half of the included papers ($n$ = 23) put forward arguments for vaccinating people in prisons, especially when vaccines were found to provide up to 95% efficacy at reducing symptomatic and severe illnesses among these individuals. The majority of these publications argued that focusing on people in prisons would form part of an optimal approach to the vaccine rollout, considering the prison population's advanced age, prevalence of multiple chronic diseases such as hypertension and diabetes, and systemic neglect prior to incarceration, all of which are known to heighten COVID-19 morbidity and mortality [12, 56, 57, 70, 74, 76].

Several papers recognised prisons as an opportunistic setting for individuals to re-engage with healthcare services, and noted that this was especially the case for people of minority ethnic groups, who are often overrepresented in prison systems [56, 67, 70, 77]. Prioritising people in prisons for COVID-19 vaccinations was seen as an important strategy for addressing health inequality, discrimination, and social exclusion, often experienced by these individuals prior to incarceration.

A number of papers recognised the status of prison staff as essential key workers and first responders [56, 76, 77]. However, vaccination of staff members alone would be unlikely to prevent COVID-19 outbreaks within prisons [36, 38, 40, 53], and therefore called for prioritisation for vaccination, in addition to other non-pharmaceutical interventions, for all those living and working in prisons [36, 38, 40, 53].

Many publications highlighted environmental factors that increase prisons' susceptibility to COVID-19, both for volume and severity of infections. These papers cited factors such as overcrowding, poor ventilation, and unsanitary conditions, which can lead congregated areas to become vectors of COVID-19 infection [51, 63, 64, 70, 74, 76, 77]. Physical distancing is almost impossible in prisons [71]. Thus, some papers argued that without vaccination it would be difficult to effectively protect those who live and work in prisons via standard prevention and control techniques, especially as inadequate testing, lack of personal protective equipment, and substandard access to healthcare remain prevalent in prisons worldwide [51, 70, 77].

Several papers suggested that COVID-19 vaccination programmes in prisons could reduce community transmission. In contrast, factors such as inter-establishment transfer, inflow and outflow of prison staff, and release of the short-term incarcerated all amplify community transmission [51, 57, 63, 70, 72]. Moreover, as prisons are often unable to offer acute healthcare, prison-based COVID-19 vaccination programmes could reduce dependency on expensive community secondary care [52].

**The extent to which people who live and work in prisons are being prioritised for COVID-19 vaccination.** The global scarcity of vaccines has forced governments to prioritise some population groups to reduce illnesses and deaths, as well as lessen pressure on healthcare systems. However, vaccine prioritisation and allocation have been managed differently

between and within countries, a situation that is politically contentious. Thirteen included papers discussed the impact of this issue on those who live and work in prisons.

Data on COVID-19 vaccination rates in prisons remains scant [60, 74]. The majority of high income countries in Europe, as well as the US, have not explicitly recognised prison populations in their national guidance and planning of COVID-19 vaccination drives [46, 51, 78]. One possible reason for this is that international guidance on vaccine rollout, such as that put forward by the European Centre for Disease Prevention and Control (ECDC), does not explicitly urge prioritisation of prison populations [71]. Indeed, women were the only group who received attention in a WHO policy document, due to their heightened risk of communicable diseases such as HIV and hepatitis B and C [77].

For countries where data are publicly available, there are notable differences in considerations given to people who live and work in prisons. In the US, COVID-19 vaccination is controlled at the state level. Some states have prioritised their imprisoned population, others have opted to only prioritise prison staff, and many have not prioritised either [46, 51, 60, 74].

On the other hand, Brazil, Kenya, Morocco, Nepal, and South Africa have explicitly prioritised people who live and work in prisons [71, 77, 79]. Moreover, a third of EU Member States, namely Austria, Croatia, Cyprus, Germany, Greece, Italy, Latvia, Portugal, Romania, and Spain have acknowledged people in prisons as a priority group in their national vaccination strategies [75, 77].

Ireland and Luxembourg recognise the vulnerable environment within which their incarcerated population is placed, given their lack of ability to self-isolate and socially distance [71, 75]. These countries stopped short, however, from nominating the incarcerated as a priority group in their national vaccination plans [71, 75]. India, Lithuania, Moldova, and Turkey chose to prioritise prison staff only, and not the incarcerated population, despite the limited efficacy of this strategy [71, 77]. Meanwhile, as of early 2021, the following countries had made no explicit prioritisation of the eligibility of people in prison and prison staff to receive a COVID-19 vaccine: Angola, Bulgaria, China, Egypt, France, Germany, Hungary, Indonesia, Norway, and the Seychelles [71]. Other nations' data remain unavailable.

Canada, the Czech Republic, Israel, Italy, Russia, Switzerland, the UK, and several US states offered COVID-19 vaccination to people in prisons based on their age and long-term health conditions, in line with prioritisation strategies for the wider community [71, 73, 77, 78]. In the UK, prison officers are not considered to have the same level of risk exposure as health and social care workers, even though they work with imprisoned individuals vulnerable to COVID-19 [66].

A modelling study from the US found that allocation of COVID-19 vaccines by age was most effective in preventing deaths [43]. On the other hand, evidence from the UK showed that restricting vaccinations to those aged 50 and above in prisons was less effective at preventing outbreaks [67]. Practically, it can be difficult to identify eligible population in prisons based on age and long-term health conditions alone due to poor data recording and limited interaction between vulnerable individuals and healthcare services [63].

**Facilitators and barriers to COVID-19 vaccination uptake for people who live and work in prisons.** Nearly four in ten publications analysed factors related to COVID-19 vaccination uptake among people who live and work in prisons. Logistically, these individuals are regarded as a captive population for which vaccinations can take place under one roof [52, 63]. This argument assumes that all individuals will provide informed consent to be vaccinated, and that healthcare services and prison authorities can coordinate to deliver the vaccination programme [46, 56, 58, 77].

Such assumptions are not always warranted. Many included papers highlighted the following barriers to vaccination uptake: high turnover in prisons and the lack of an efficient system

to track individuals moving in and out of prisons, limiting ability to complete the two vaccine schedule [56, 59, 61, 76, 77]. Furthermore, poor planning resulted in vaccines going to waste as they could not be used elsewhere in a timely way [52, 63]. Also, prisons have historically had issues with not having sufficient staff to administer vaccinations. Staff being forced to self-isolate further aggravates existing understaffing problems [61, 67].

Two US studies highlighted the issue of the lack of critical infrastructures such as cold-chain support to deliver the vaccine at recommended low temperatures, or sufficient capacity to administer two COVID-19 vaccines at the correct intervals [46, 56]. Although unique to the US, one publication highlighted issues of affordability, noting differences in expectation of whether individuals' health insurance, health departments, or prison authorities should pay for vaccines [61].

Several US papers analysed how the exclusion of people in prisons from COVID-19 vaccination trials prevented them from having early access to vaccines. Exclusion was based on various factors including increased vulnerability, ethical and recruitment challenges, and lack of participation in design of studies [47, 55, 57]. Several papers raised the ethical arguments of need for informed consent, avoiding coercion, and minimising harm, recommending that oversight boards monitor trials and obtain input from these populations to enable their participation [47, 48]. Publications raised further relevant issues for the prison population, including real and perceived pressure from authorities to participate in trials, pressure to participate to receive the care they cannot afford, uncertain risks of vaccines being trialled, limited or non-existent on-site clinical support, and the absence of an agreement that people in prisons would benefit from the vaccines if they were found to be safe and effective [47, 62].

**Vaccine hesitancy.** Vaccine hesitancy among incarcerated individuals, as with the general population, is an emerging topic in the extant COVID-19 vaccination literature. The 11 studies that discussed this issue were all published in either the UK or the US. The majority of papers described high rates of vaccine hesitancy among people who live and work in prisons [37, 39, 46, 51, 54, 80]. However, two studies from England provided conflicting data, with one finding that nearly eight in ten people in prison would accept a vaccine if they were offered one [63], while another, using data from five prisons in England and Wales, found that between a third and two thirds of people in prisons declined a first vaccine dose [80]. Regarding demographic correlates of vaccine hesitancy, several publications stated that younger people and those from ethnic minorities were less likely to opt for vaccines [37, 39, 54, 63].

Reasons for vaccine hesitancy among incarcerated individuals appear to be similar to those reported in the general public. These include concerns about safety and efficacy, and around the rapid speed that vaccines had been developed [39, 63]. A lack of educational materials about COVID-19 vaccines amplified these concerns, subsequently leading to endorsement of conspiracy theories and feelings of apathy or general scepticism towards vaccines [39, 49, 51, 54, 80].

Suspicion and distrust of authorities were also cited as key reasons for refusing vaccines, motivated by longstanding abuse and violation of human rights experienced by individuals in criminal justice institutions [37, 39, 49, 51, 54]. Mental health issues could exaggerate these issues [58]. For staff, a lack of appreciation of the link between prison and community transmission was found to be a key factor explaining vaccine hesitancy [46].

Nevertheless, some publications found that initially hesitant individuals were willing to learn more about vaccine safety and efficacy [37, 54]. One US study found that a substantial proportion of residents who had initially declined a first dose accepted a later offer, an important indication that hesitancy is not fixed [54]. As such, a small number of publications suggested improving health literacy strategies and using these alongside peer-to-peer persuasion techniques to address weaknesses in health education and reduce mistrust and misinformation

[37, 45, 49, 61]. One US paper considered how crisis management is influenced by knowledge, attitudes, and practices, and proposed that simultaneous vaccination of incarcerated individuals and staff could reduce mistrust among those who live in prisons [63].

**Vaccine populism.**   Three included publications addressed vaccine populism. Discussion focused on political arguments in Canada, Israel, South Africa, and the United States, countries where politicians have influenced vaccination prioritisation by claiming that people in prison are less deserving of vaccines than healthcare workers and older people, and ignoring epidemiological evidence of potential harm to the wider population [51, 79]. In Israel, prison authorities' refusal to vaccinate people in prisons caused the country's Attorney General to have to direct the prison service to do so [73]. Collectively, these papers argue for a paradigm shift whereby COVID-19 vaccination for those in prison is viewed as benefiting the whole community and not just incarcerated individuals. The papers also stress the need for governments to use evidence and to more effectively communicate their decision-making to appease the media and the public [58, 73].

**Access to vaccines as a human right.**   Four publications framed access to COVID-19 vaccines by people who live and work in prisons in terms of medical ethics and legal arguments. These papers highlighted how members of the public often view people in prisons unfavourably or as unworthy, similar to the populist arguments of worthiness made by some politicians [72, 76]. The papers suggest as a starting point that all humans have equal moral worth irrespective of their culpability [72, 76]. As people from ethnic minorities and poorer social and economic backgrounds are overrepresented in prison populations, prioritising them as vaccine candidates would improve their access to healthcare services [52, 58].

Beyond medical ethics positions, some papers engaged with a duty of care argument, that people who have been deprived of liberty are the responsibility of the state [12, 63]. As such, the principle of equivalence applies, whereby these people should be entitled to access the same standards of healthcare available in the community, free of charge and free of discrimination by legal status [12, 46, 63]. Two publications, drawing upon experiences in the US and Brazil, propose a last resort of engagement with the court system to intervene in vaccine allocation prioritisation on the basis of incarcerated people's constitutional rights [52, 68].

**Vaccination should run in parallel with other mitigation actions.**   While vaccination is key for mitigating the risk of COVID-19 transmission in prisons, other interventions are required to run in parallel. Of six papers that covered this theme, the majority suggested reducing the prison population to reduce overcrowding, known to be a key vector of COVID-19 transmission in prisons [42, 44, 50, 59]. Papers also argued that governments must continue to implement adequate mitigation strategies involving mass testing, sufficient PPE, and quarantine [41, 50, 59, 69]. Limiting transfers between establishments and improving ventilation systems in prisons can help to mitigate the spread of COVID-19 prisons [50].

## Discussion

Evidence from the present review suggests that people who live and work in prisons have generally not been prioritised for COVID-19 vaccination programmes, notwithstanding an awareness of the vulnerability for these populations. This conclusion, to a large extent, discounts the majority of low and middle income countries, given the absence of published vaccination strategies from these countries.

This issue has been exacerbated by political and operational barriers affecting vaccination rollout, and by individual-level issues such as vaccine hesitancy. Ethical and legal arguments on health equity, inequality, the principle of equivalence, and governments' duty of care provide a backdrop for our analysis.

Many of the included studies used health inequalities as justification for vaccinating people who live and work in prisons. Overwhelming and conclusive evidence that people in prisons experience greater burden of disease than the general population [3, 4], higher levels of pre-existing chronic health conditions, and systemic neglect prior to incarceration justify incarcerated populations being prioritised in national vaccination programmes [12, 56, 57, 70, 74]. There was also recognition of the potential utility of prisons, such that a large number of vulnerable individuals could be vaccinated in one setting [56, 67, 70, 77]. Prioritising people who live and work in prisons could help address the marginalisation and exclusion experienced prior to incarceration, protect these individuals from hospitalisation and death, and prevent the further spread of COVID-19. Equally, prison is an opportunistic setting to improve vaccine coverage in the whole population, especially when people in prisons often do not reach out to community health services despite universal health coverage. Several studies notably framed prison staff as first responders and essential key workers, similar to health and social care staff, who are exposed to higher risk of COVID-19 [18, 56, 76, 77]. Prioritising vaccination for these workers would help to ensure that absenteeism does not impact the effectiveness of prison services [65].

Included studies also emphasised the role of environmental factors amplifying the spread of COVID-19 in prisons. These range from overcrowding and unsanitary prison cells to a lack of testing facilities or adequate protective equipment [51, 57, 63, 70, 71, 74, 76, 77]. These findings are in line with reports that have referred to prisons as 'epidemiological pumps', meaning that the environment of people who live and work in prisons increases susceptibility to COVID-19 infection [6]. Prison health is public health, and widespread COVID-19 vaccination in prisons would reduce community transmission stemming from inflow and outflow of staff and the short-term incarcerated [51, 57, 63, 70, 72]. It would also mitigate dependencies on secondary care in the community at facilities that are often financially and operationally stretched [52]. The whole of society stands to benefit from improved vaccination uptake by imprisoned people and prison staff.

Clear evidence of the vulnerability of people who live and work in prisons has unfortunately not translated into them being prioritised in most countries' COVID-19 vaccination programmes. That said, such gaps in national vaccination programmes can only be observed and addressed if robust data are available. In many cases, such data do not yet exist [60, 74]. Additionally, vaccination registration could be incorporated into the national immunisation information system, which could then inform future assessment of vaccination coverage in prisons.

Our review discovered that only a minority of countries–specifically Brazil, Kenya, Morocco, Nepal, and South Africa–have prioritised people who live and work in prisons for vaccination [71, 75, 77, 79]. Many countries–specifically Austria, Croatia, Cyprus, Germany, Greece, Italy, Latvia, Portugal, Romania, and Spain–have not explicitly prioritised these individuals as part of their vaccine rollout, even though some unambiguously recognised the vulnerability of the prison population in official publications [71, 75, 77]. Such inconsistencies are irrational, unethical and discriminatory, and perpetuate the marginalisation and social exclusion that many imprisoned people experienced both before and during their incarceration. Furthermore, addressing the health needs of imprisoned people is likely to have a positive impact on reducing reoffending, thus benefiting the whole of society.

We found that a small number of countries–specifically Canada, the Czech Republic, Israel, Italy, Russia, Switzerland, the UK, and several US States–have prioritised people in prisons for vaccination based on their age and if they suffer from a chronic illness, in line with recommendations for the general public [71, 73, 77, 78]. There was a lack of evidence to demonstrate the value of such prioritisation. One study in the US found that age prioritisation could reduce death rates in prisons [43], and one UK study found it to have little effectiveness in preventing

COVID-19 outbreaks [67]. This approach runs the risk of ignoring physical and environmental risk factors, or the high rates of comorbidities experienced by people in prison, with existing research on older people in prisons calling for an eligibility age of fifty. It can also be difficult to identify eligible populations in prisons as a result of poor data recording and limited previous interaction with healthcare services [63].

Prioritisation of those in prison for vaccination has also been limited by some politicians' moral relativeness and comparing the worthiness of people in prison with that of other community groups. A small number of publications discussed this issue [51, 79]. A policy of limiting care for those in prison is in line with the principle of less eligibility [81], which negates the fundamental right of imprisoned populations to receive the same level of healthcare as the free population.

Many of the included papers considered ethical arguments. Access to vaccines is a human right, and prison populations should be carefully considered as candidates for COVID-19 vaccine trials. One ethical argument put forward was that humans have equal moral worth, irrespective of their culpability [72, 76]. A second was that prioritising people in prison for vaccination can actually reduce the impact of health inequalities, particularly as prison populations have an overrepresentation of people from ethnic minorities and from lower socioeconomic background [5, 52]. A third argument framed states' duty of care towards those deprived of liberty using the principle of equivalence, i.e., that these populations are entitled to access the same standards of healthcare available in the community [12, 46, 63].

Our analysis of official documents identified an absence of any unified approach regarding prioritisation of people who live and work in prisons, meaning that there is no clear framework for nations to adopt. The European Centre for Disease Prevention and Control does not explicitly urge prioritisation of people who live and work in prisons [71]. In contrast, the WHO advocated for a more inclusive approach by advocating people living and working in detention facilities should be included in national COVID-19 vaccination plans [77]. The lack of synchronisation across the international health guidance does not conform to the Mandela Rules, the Bangkok Rules, the Beijing Rules, or the Havana Rules, or to the International Covenant on Economic, Social and Cultural Rights, which asserts that governments have a duty to protect those in their care when states deprive their liberty. Furthermore, the UN's Sustainable Development Goals 2030 cannot be reached if the prison population is "left behind" regarding COVID-19 vaccination [31].

Prior to COVID-19, there was a limited body of research on vaccine hesitance among people in prisons compared to the general population. Yet, our analysis illustrates high rates of COVID-19 vaccine hesitancy among people who live and work in prisons, in line with evidence from the wider population. Hesitancy is higher in incarcerated individuals who are young and who come from a minority ethnic background [37, 39, 54, 63]. Reasons for hesitancy commonly included concerns around vaccine safety and efficacy, often fuelled by individual apathy, scepticism, and belief in conspiracy theories [39, 49, 51, 54, 63, 80]. Studies noted that suspicion and distrust among the incarcerated were likely related to long-standing abuse experienced during their time in the criminal justice system [37, 39, 49, 51, 54].

Nonetheless, there is some hope that such hesitancy is fluid. Studies reported that many hesitant individuals were willing to receive information and have their concerns around vaccine safety and efficacy assuaged [37, 54], and a US study found that a substantial proportion of residents who had declined a first vaccine dose accepted a later offer [54].

Despite the utility of targeting prisons as a setting for COVID-19 vaccination rollout, our analysis demonstrates how logistical barriers can hamper such a policy. The need for cold-chain support to deliver the vaccine at recommended low temperatures, and the necessity of administering vaccinations twice at recommended intervals remain the key challenges [46,

56]. High turnover of people in prisons and difficulty tracing those that return to the community can compound this issue, as can vaccine wastage, whereby vaccines cannot be used in another setting [52, 56, 59, 61, 63, 76, 77]. These findings are in line with earlier studies that illustrate historical issues of underfunding [17, 31]. Economic limitations can limit the government's ability to purchase vaccines, which means fewer doses are available for people in prisons, and at the same time leading to prisons not having sufficient staff to administer vaccinations, and increased uncertainty when staff are required to self-isolate [61, 67].

Our analysis revealed that the majority of papers on vaccination trials were opposed to the inclusion of people in prisons. Objections included the real and perceived pressure from authorities to participate, demands to participate to receive care that could otherwise not be afforded, risks associated with the vaccines being trialled, limited or no on-site clinical support for participants, and no guarantee that participants would benefit from the vaccines if they were found to be safe and effective [47, 62]. These concerns tended to trump the perceived benefits of participation. Indeed, many publications raised the issue that recruiting trial participants from the prison population would reinforce the perpetual exploitation, abuse, and neglect of people in prison. Other studies suggested precursors for participation, such as closer scrutiny by an oversight board and greater consideration of the lived experiences of incarcerated people [47, 48].

Finally, a limited number of studies acknowledged that vaccination is not a panacea. Virus mutation, lag in vaccination uptake, and vaccine hesitancy mean that non-pharmacological interventions must run in tandem with vaccination for the foreseeable future, a fact that is well-documented. Studies put forward interventions such as reducing the prison population, an ongoing commitment to provide adequate testing and personal protection equipment in prisons, limiting transfers between establishments, and improving prison environments [41, 42, 50, 59, 69]. Implementing any of these interventions would not only be in line with existing government obligations regarding prison health, but would also protect vulnerable populations in a precarious environment. Vaccinating the prison population could enable the relaxation of restrictions on movement, meaning that in-person visitation and prison activities could resume safely, while maintaining an appropriate testing regime.

## Way forward

Our review has identified several gaps in policy and research related to COVID-19 vaccination prioritisation among people who live and work in prisons.

Based on our analysis, we urge politicians, policymakers, academics and practitioners to consider prioritising people who live and work in prisons as part of the global COVID-19 vaccination strategy. National strategic and operational plans should be underpinned by intersectional and inter-sectoral approaches, and should aim to address the continued marginalisation, discrimination, and exclusion of people in prison.

Prioritisation, rather than mere inclusion, of people who live and work in prisons, is critical. Evidence on the morbidity and mortality of this population is clear. Due to their high-risk environment, those in prisons deserve a high ranking on national vaccination plans. Vaccination programmes are not just for the benefit of individuals and prison establishments, but for the whole community. Prison walls are permeable and vaccinations in these settings will prevent community transmission through staff, visitors, or residents returning to their communities.

Due to sparse data on much of the global vaccination programme, international organisations such as the WHO, should champion the use of a minimum dataset to capture vaccination data. Transparent data collected through a robust, verified, and validated system will allow

timely monitoring and encourage accountability for poor observance by government authorities, courts, and civil society organisations.

Much more effort should be made to reduce the prison population. Doing so would reduce the elevated risk of COVID-19 caused by overcrowding, reduce reliance on imprisonment as a criminal justice measure, and help to address the historical underfunding of prisons that continues today.

It is also important that public health advocates and others continue to challenge the vaccine populism perpetuated by politicians and the media. This should involve informing the population of the impact of vaccination programmes not reaching prisons on community health, calling upon politicians and the media to exercise restraint when debating this contentious matter to avoid polarising 'us against them' narratives, and encourage debate that is guided by scientific evidence rather than political rhetoric. It is also the responsibility of governments to make and communicate their decisions transparently, and to convey to the media and the public of the utility of vaccinating prioritised communities, including people in prison.

People in prisons should be educated about the safety and efficacy of vaccines to address vaccine hesitancy and promote health literacy. Doing so could allay concerns and prevent the spread of misinformation, mistrust, and conspiracy theories. Healthcare staff should, as much as possible, act independently from prison authorities as is recommended in WHO guidelines to promote trust in the vaccination information they provide. Peer educators, as well as family and friends, can play a role in educating vaccine hesitant individuals about COVID-19 vaccination.

As a last resort, judicial rulings can be called upon to accelerate prioritisation of vaccination for people who live and work in prisons. Indeed, American and Brazilian experiences show how the judiciary can intervene in vaccine allocation prioritisation on the basis of enforcing the constitutional rights of incarcerated people [52, 68]. These experiences can be extrapolated elsewhere to allow for greater parity of treatment for marginalised populations.

Finally, more empirical research is urgently needed to gain in-depth understanding of COVID-19 vaccination planning, delivery, and acceptability in prisons. While the majority of empirical research included in this review was of good quality, the current paucity of such research meant that we had to rely heavily on opinion-based publications. We also note the low number of qualitative studies on this topic. Such studies would have enabled insights into the 'why' and 'how' of these issues. COVID-19 cases and vaccination rates are also poorly recorded in staff working in prisons. We recommend a move towards empirical studies to enrich the evidence base on this topic. Equally, findings from these studies could be used as a lever for the future expansion of vaccine offers in prisons beyond COVID-19.

## Strengths and limitations of the study

To our knowledge, this is the first scoping review of the prioritisation of COVID-19 vaccination for people who live and work in prisons. Its rigour and transparency are illustrated through a comprehensive search strategy, which facilitated identification of a considerable number of articles of various study design and publication type. Following PRISMA guidelines, we charted and analysed selected studies and undertook a quality assessment as part of our quality control process. Our analysis informed a set of clear recommendations for policy and research.

Most of the included studies were conducted in the United States, and when we included studies from other nations, countries from the Global North dominated our analysis. Due to geopolitical differences, generalisability of our analysis may therefore be limited to these

regions. Thus, our study calls for additional empirical studies from the Global South to build a more holistic picture of vaccination in prisons across the world.

Finally, COVID-19 vaccination is of course a dynamic topic. While every effort has been made to ensure that the information contained in this review is correct as of the time of writing, some countries may have since changed their policies. Nevertheless, our review provides the first insights into the topic, and we hope that future studies will build upon these findings.

## Conclusion

Our analysis of the published literature on COVID-19 vaccinations illustrates that people who live and work in prisons are not being prioritised for COVID-19 vaccinations despite a recognition of their higher rates of morbidity and mortality.

This issue is compounded by factors such as vaccine populism, often perpetuated by politicians, lack of advocacy by key health organisations, vaccine hesitancy, and logistical issues affecting prison establishments. The high quality of papers included for our assessment supports the strength of our findings but they preclude generalisability given that the main focus of these publications was on high income countries.

Evidence presented in this review shows that governments must prioritise prison populations in national vaccination programmes, and that organisations such as the WHO should request transparent data so that rollout and uptake of vaccines in these populations can be adequately monitored. Additionally, efforts should be geared towards reducing the prison population, challenging vaccine populism, and improving health literacy among people who live and work in prisons to increase vaccination uptake. Court interventions could be used as a last resort. Further empirical studies on COVID-19 vaccination among people who live and work in prisons are also needed to enrich discussions on this topic and operationalise our responses beyond the current pandemic.

In December 2021, the WHO predicted the demise of COVID-19 in 2022 [82]. People who live and work in prisons clearly have a stake in achieving this vision. Treating vaccines as public goods and prioritising high-risk groups in the race against the virus is necessary. For the prison population, the adage that "no one is safe until everyone is safe" certainly rings true.

## Supporting information

**S1 Appendix. Subject headings.**
(DOCX)

**S2 Appendix. Abbreviated data charting form.**
(DOCX)

**S3 Appendix. Quality assessments of the included studies.**
(DOCX)

**S1 Checklist. Preferred Reporting Items for Systematic reviews and Meta-Analyses extension for Scoping Reviews (PRISMA-ScR) checklist.**
(DOCX)

## Author Contributions

**Conceptualization:** Nasrul Ismail, Lara Tavoschi, Babak Moazen, Alicia Roselló, Emma Plugge.

**Formal analysis:** Nasrul Ismail, Emma Plugge.

**Investigation:** Nasrul Ismail, Lara Tavoschi, Babak Moazen, Alicia Roselló, Emma Plugge.

**Methodology:** Nasrul Ismail, Lara Tavoschi, Babak Moazen, Alicia Roselló, Emma Plugge.

**Project administration:** Nasrul Ismail, Emma Plugge.

**Resources:** Nasrul Ismail.

**Validation:** Nasrul Ismail, Lara Tavoschi, Babak Moazen, Alicia Roselló, Emma Plugge.

**Visualization:** Nasrul Ismail, Emma Plugge.

**Writing – original draft:** Nasrul Ismail, Lara Tavoschi, Babak Moazen, Alicia Roselló, Emma Plugge.

**Writing – review & editing:** Nasrul Ismail, Lara Tavoschi, Babak Moazen, Alicia Roselló, Emma Plugge.

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
