## [Decision Letter · Decision Letter 0]

9 May 2022

PONE-D-22-09409COVID-19 vaccine for people who live and work in prisons worldwide: A scoping review.PLOS ONE

Dear,

Thank you for submitting your manuscript to PLOS ONE. After careful consideration, we feel that it has merit but does not fully meet PLOS ONE’s publication criteria as it currently stands. Therefore, we invite you to submit a revised version of the manuscript that addresses the points raised during the review process. Please submit your revised manuscript by 23 June 2022. If you will need more time than this to complete your revisions, please reply to this message or contact the journal office at plosone@plos.org. Please include the following items when submitting your revised manuscript:A rebuttal letter that responds to each point raised by the academic editor and reviewer(s). You should upload this letter as a separate file labeled 'Response to Reviewers'.A marked-up copy of your manuscript that highlights changes made to the original version. You should upload this as a separate file labeled 'Revised Manuscript with Track Changes'.An unmarked version of your revised paper without tracked changes. You should upload this as a separate file labeled 'Manuscript'.

We look forward to receiving your revised manuscript.

Kind regards,

Muhammad Shahzad Aslam, Ph.D.,M.Phil., Pharm-D

Academic Editor

PLOS ONE

Journal Requirements:

Reviewers' comments:

Reviewer's Responses to Questions

**Comments to the Author**

1. Is the manuscript technically sound, and do the data support the conclusions?

Reviewer #1: Partly

Reviewer #2: Yes

2. Has the statistical analysis been performed appropriately and rigorously? 

Reviewer #1: N/A

Reviewer #2: Yes

3. Have the authors made all data underlying the findings in their manuscript fully available?

Reviewer #1: Yes

Reviewer #2: Yes

4. Is the manuscript presented in an intelligible fashion and written in standard English?

Reviewer #1: Yes

Reviewer #2: Yes

5. Review Comments to the Author

Reviewer #1: This is an important study, and while I checked the box for "major revisions" (rather than "minor revisions" as there wasn't a "some revisions" option), I don't actually think the changes will be a huge lift, so don't despair!

1. Separate out the review findings from the commentary. Right now, the article mixes up the scoping review results, the findings and the analysis, almost like an opinion column mixed up with a scoping review.

The review findings should be presented objectively. The analysis and discussion should follow the findings and be clearly distinct in a separate section. Here is one example:

"Our review discovered that only a minority of countries have prioritised people who

live and work in prisons for vaccination.71,75,77,79 Many countries have not explicitly

prioritised these individuals as part of their vaccine rollout, even though some

unambiguously recognised the vulnerability of the prison population in official

publications.71,75,77 Such inconsistencies are irrational, unethical and discriminatory,

and perpetuate the marginalisation and social exclusion that many imprisoned

people experienced prior to and during their incarceration. Furthermore, addressing

the health needs of imprisoned people is likely to have a positive impact on reducing

reoffending, thus benefiting all of society."

Everything that follows "Such inconsistencies..." belongs in a separate discussion section that you put after the review findings. This kind of change needs to be made throughout, but especially from pages 18 on.

2. Tone down the conclusions to align with your actual findings. Right now, the conclusions begin with a very strong statement: "People who live and work in prisons are not being prioritised for Covid-19 vaccinations"... Actually that is not exactly what the findings state. It sounds more like you found from the scoping review that there's a lack of data (especially from LMIC), that a small number of countries are prioritizing poeple in prisons, a larger number of countries are not, and that there's a lack of data from many others. Your conclusions will be more compelling if they are clearly and specifically aligned to the findings.

3. How many countries are we talking about? It's really unclear from the text when you say "a small number of countries" have done something or "a minority of countries" or "some countries" how many we mean. As this is an important point of the article, could you create a table showing which countries have done what and where we have no data? That would make it easier to see where things stand.

Overall this is important work, and especially important that you link the rights of prisoners, rights of people working in prisons, and rights of marginalized communities. I hope you will be able to turn this around quickly for publication. It might even be worthwhile writing a separate opinion column or editorial where you can really cut loose, freely denounce the human rights violations revealed by the study and not be constrained by the strict format of a scoping review.

Reviewer #2: 1. The conclusion is in accordance with the data presented. But not yet connected with research questions (page 5) and research objectives (page 4). the objectives of the research is not clear enough or there is not enough explanation. In my view, the purpose of this study is to look at the quality of the literature, not to convey information about COVID-19 vaccination in prisons from the literature reviewed.

2. Data analysis is appropriate.

3. The manuscript already has data that provides the required information

4. the manuscripts are presented in an intelligible fashion

6. PLOS authors have the option to publish the peer review history of their article (what does this mean?). If published, this will include your full peer review and any attached files.

Reviewer #1: **Yes: **Sara (Meg) Davis

Reviewer #2: **Yes: **Nisaa Nur Alam

---

## [Author Response · Author response to Decision Letter 0]

14 Jul 2022

On behalf of all the authors, we would like to thank you for your recent correspondence regarding the initial decision on the manuscript entitled “COVID-19 vaccine for people who live and work in prisons worldwide: A scoping review.” Please see the attached letter with our responses to all queries raised by the reviewers. Thank you.

---

## [Decision Letter · Decision Letter 1]

4 Aug 2022

COVID-19 vaccine for people who live and work in prisons worldwide: A scoping review.

PONE-D-22-09409R1

Dear,

We’re pleased to inform you that your manuscript has been judged scientifically suitable for publication and will be formally accepted for publication once it meets all outstanding technical requirements.

Kind regards,

Muhammad Shahzad Aslam, Ph.D.,M.Phil., Pharm-D

Academic Editor

PLOS ONE

Additional Editor Comments (optional):

Reviewers' comments:

Reviewer's Responses to Questions

**Comments to the Author**

1. If the authors have adequately addressed your comments raised in a previous round of review and you feel that this manuscript is now acceptable for publication, you may indicate that here to bypass the “Comments to the Author” section, enter your conflict of interest statement in the “Confidential to Editor” section, and submit your "Accept" recommendation.

Reviewer #1: All comments have been addressed

2. Is the manuscript technically sound, and do the data support the conclusions?

Reviewer #1: Yes

3. Has the statistical analysis been performed appropriately and rigorously? 

Reviewer #1: N/A

4. Have the authors made all data underlying the findings in their manuscript fully available?

Reviewer #1: Yes

5. Is the manuscript presented in an intelligible fashion and written in standard English?

Reviewer #1: Yes

6. Review Comments to the Author

Reviewer #1: (No Response)

7. PLOS authors have the option to publish the peer review history of their article (what does this mean?). If published, this will include your full peer review and any attached files.

Reviewer #1: **Yes: **Sara L.M. Davis

---

## [Editor Report · Acceptance letter]

1 Sep 2022

PONE-D-22-09409R1 

COVID-19 vaccine for people who live and work in prisons worldwide: A scoping review 

Dear Dr. Plugge:

I'm pleased to inform you that your manuscript has been deemed suitable for publication in PLOS ONE. Congratulations! Your manuscript is now with our production department. 

Kind regards, 

on behalf of

Dr. Muhammad Shahzad Aslam 

Academic Editor

PLOS ONE